# Landscape and Socioeconomic Factors Determine Malaria Incidence in Tropical Forest Countries

**DOI:** 10.3390/ijerph21050576

**Published:** 2024-04-30

**Authors:** Allison Bailey, Paula R. Prist

**Affiliations:** 1EcoHealth Alliance, 520 Eighth Ave., Ste. 1200, New York, NY 10018, USA; prist@ecohealthalliance.org; 2Future Earth, One Health, 413 Chukar Ct., Fort Collins, CO 80526, USA

**Keywords:** malaria, environment, deforestation, land-use change, global

## Abstract

Deforestation, landscape dynamics, and socioeconomic factors within the tropical Americas, Africa, and Asia may have different impacts on malaria incidence. To evaluate how these drivers affect malaria incidence at the global and regional scale, we collected malaria incidence rates from 2000 to 2019 from 67 tropical countries, along with forest loss, land use change types, and socioeconomic elements. LASSO regression, linear mixed effect modeling, and k-fold cross validation were used to create and evaluate the models. Regionality plays a role in the significance of varying risk factors. The Tropical Americas model had the highest coefficient of determination (marginal R^2^ = 0.369), while the Africa model showed the highest predictive accuracy with only a 17.4% error rate. Strong associations between tree cover loss (β = −4037.73, *p* < 0.001) and percentage forest area (β = 5373.18, *p* = 0.012) in Africa, and percent of key biodiversity areas under protection (β = 496.71, *p* < 0.001; β = 1679.20, *p* < 0.001) in the tropical Americas and Asia with malaria incidence indicates that malaria risk should be considered during conservation policy development, and recommends that individual approaches to policy and investment be considered when implementing malaria interventions on different spatial scales.

## 1. Introduction

In 2022, 249 million cases of malaria were reported in 85 malaria-endemic countries [1]; malaria is a vector-borne, protozoan infectious disease transmitted by the bite of the *Anopheles* sp. female mosquito. Two species compose 95% of all infections: *Plasmodium falciparum* and *Plasmodium vivax*, with the former being the most dominant and causing the greatest number of illnesses and deaths, while the latter is often considered less severe yet can increase morbidity with recurrent infections [2,3,4,5]. Due to the vector’s preferred environment of warm temperatures and increased precipitation and humidity, malaria is most common in the tropical regions of the world, particularly throughout Africa, Southeast Asia, and the Amazonian Basin [2,3,6]. Despite a global increase in malaria cases from 2019 to 2020 [6] and 2019 to 2022 [1], many regions of the world have seen a decline or elimination of malaria due to concentrated efforts from global humanitarian organizations and national governments through acts of prevention and treatment (antimalarial drugs, intravenous medications, vector control, insecticide-treated netting, indoor residual spraying) [2,4,5] and health investment [6,7,8].

However, as measures are taken to control the most common parasitic disease in the world, the landscapes that contain the infectious *Plasmodium* are rapidly shifting. Global forest loss has approximated 4.7 million hectares per year since 2010 [9,10,11,12], with tropical forests even more at risk. Countries such as Indonesia, Malaysia, Paraguay, Bolivia, Zambia, and Angola show an increase in deforestation and a gross loss of 7.3 Mha per year (yr-1) between 2000 and 2018 [10,13]. The reasons for deforestation are vast and varied, dependent on the socioeconomics, climate, and resources of the region, and often pressured by importation demands from G7 countries and China [14]. Logging and other forestry products are the primary drivers of forest degradation in Latin America and Asia (more than 70%), while charcoal production and firewood collection are the prevailing reasons in Africa (approximately 50%) [15], with forestry products globally totaling 0.8Mha loss per year [16].

More destructive is the converting of native or primary forests to pasture, farmlands, or urban landscapes; between 2005 and 2015, 62% (5.5 Mha per year) of forest loss was attributed to expanding commercial croplands, pastures, and tree plantations [16]. Brazil and Indonesia accounted for 44% of deforestation due to land-use change, followed by Argentina (7%) and Paraguay (4%) [16]. In Latin America between 2002 and 2015, expanding pastureland for cattle ranching resulted in a 2.2 MHa loss of forestland per year [16,17,18], while commercial agriculture (68% of deforestation) from soybeans resulted in approximately 0.4 MHa [15,16]. In Asia, deforestation is the result of logging and land-conversion to plantations, which is estimated to be the direct cause of 3 to 50% of regional tropical deforestation [18,19]. Africa’s tropical forests are not deforested at the rates of the tropical Americas or Asia due largely to land-use change occurring from subsistence and small-scale farming, yet commercial agriculture and logging are on the rise in the Congo Basin [15,18].

The relationship between malaria risk and deforestation or land-use change is still contradictory. It has been hypothesized that deforestation can alter water availability and pooling, as well as surface temperatures, which can affect the incidence of larval and adult mosquitoes and increase malaria risk [3,20,21,22,23]. However, other studies have found opposing results. The majority of regional- or national-level studies in Latin America have found a linked association to increased malaria or malaria risk, either directly through deforestation or from changes in land use or land cover from a forest landscape [24,25,26,27,28,29,30]. In Africa, results are often more mixed due to complicating factors of socioeconomics, endemicity of mosquitoes, and access to malaria treatment and prevention: it is hypothesized that smallholder agriculture expansion into endemic regions may not show as intense of a relationship to malaria risk as countries that experience rapid expansion into in low endemicity, while other studies acknowledge that consistent access to health plays a significant role in malaria cases [31,32,33,34,35,36,37,38,39]; the region deserves closer attention due to impacts from having the highest incidence of malaria in the world [6]. Like Africa, research results on deforestation and malaria in Asia are more mixed [40,41,42,43,44]. Although papers have identified increased malaria risk with palm oil plantations [45,46], others have determined that malaria risk from deforestation is varying and at its highest during active deforestation and decreases as the land becomes more pastural [44,47]; this same pattern is also found in other regions of the globe such as the Amazon [29]. Global research highlights that malaria risk linked to deforestation is highly dependent on regional factors such as elevation and water coverage, type of land use change, mosquito species, and forest type [21,48,49].

Wide-reaching trends of deforestation and land use change on malaria risk are unknown and well-needed. Although previous studies have found direct, singular links between factors such as land use, health expenditures, or government aspects, few have tried to link these multiple characteristics into one cohesive study design. Our aim is to examine the current trend of malaria incidence and the global malaria risk in response to forest loss and land use change, in addition to other socioeconomic and economic features, by measuring country-level malaria for 67 countries with tropical forests. We further examined if regionality contributes to a significant role in the differing rates of malaria incidence beyond the global level and if drivers of malaria risk vary according to each region. Our results can be used to increase the understanding of which factors most affect malaria risk in the different regions of the world, to understand their contribution, and thus to be aware of the potential for new risks under a changing planet.

## 2. Materials and Methods

Three selection criteria were employed in the country collection of the global dataset for our study: (1) the country must have incidence of endemic malaria of either *Plasmodium falciparum* or *Plasmodium vivax* in 2019; (2) the country must include any amount of tropical forest in its territory; and (3) the country must have data available in the Global Health Data Exchange (GHDx) database (https://ghdx.healthdata.org/, accessed on 18 May 2023). A total of 86 countries were identified as having an incidence of malaria using visual identification from the Malaria Atlas Project (https://malariaatlas.org/, accessed on24 April 2023). A list of countries with tropical forests was generated using the criteria laid out by Grainger [50] after exploring how inconsistencies in estimates of tropical forest areas from the Forest Resource Assessments of the United Nations Food and Agriculture Organization have led to widely varying numbers of countries identified with tropical forests. For this study, we used Grainger’s [50] listed 90 countries with tropical forests. For the final global dataset, 71 countries fit the criteria above; countries were removed for either having an incidence of malaria but not containing tropical forests or housing tropical forests but either had no endemic incidence of malaria or malaria was eliminated from the country. Countries were then grouped according to UN M49, or the Standard Country or Areas Codes for Statistical Use, developed and maintained by the United Nations Statistics Division for statistical groupings [51]; exceptions were Papua New Guinea, which was moved to Southeastern Asia instead of its official classification of Melanesia, and Sudan, which was moved to Eastern Africa instead of its official classification of Northern Africa, so to prevent two regional groupings with a single country.

Variables of interest to our study were collected from online databases (Table 1). For each country, malaria incidence per 100,000 population was collected from 2000 to 2019 for a total of 20 years. Malaria incidence from the online database was defined as the number of new cases in a year divided by the mid-year population size. The data period and analysis exclude the majority of the SARS-CoV-2 pandemic, and therefore, the reported malaria incidence is not believed to have been impacted by the public health complications of that time. Several datasets related to land change measurements, socioeconomic status of country citizens, and national economic profiles were selected after literature exploration for possible influence on malaria incidence in the countries of interest. Variables were included in the final dataset if they contained enough yearly data (at minimum 15 years of data) or if the variables were static or linear enough to reasonably interpolate missing data. Not one singular variable was selected to represent deforestation and land use change: tree cover loss for forests of 30% or greater canopy cover, forest area percent of total land area, and agricultural area percent of total land area capture aspects of land use change in countries of interest. In addition, variables average proportion of terrestrial key biodiversity areas (KBAs) covered by protected areas and total official development assistance for biodiversity by recipient countries may indicate the quality of forest and other terrestrial environments in the country as a result of government priorities on protecting important areas of biodiversity. Total malaria spending per person, universal health coverage index, and net official development assistance received as a percent of a country’s gross national income (GNI) were variables used to represent a country’s efforts in healthcare and malaria prevention, as well as citizen welfare. All variables were extracted at the country-level scale. Finally, the economic variables of agricultural, forestry, and fishing as a percent of gross domestic product (GDP), total natural resource rent as a percent of GDP, and GDP per capita were selected to indicate a country’s economic status and industries related to forest presence or quality.

When applicable, variables were converted to SI units. For variables that included missing data, one of three methods were applied: (1) if yearly data at the beginning of the time series were missing, values were replaced using “first observation carried backward”; (2) if yearly data at the end of the time series were missing, values were replaced using “last observation carried forward”; (3) if yearly data in the middle of the time series were missing, values were replaced using linear interpolation; all methods were conducted with the R package “zoo” [63]. Due to excessive missing data in multiple independent variable datasets that could not be addressed through interpolation or other missing-data correction methods, the countries Cabo Verde, Somalia, South Sudan, and Venezuela were removed; the final dataset for analysis contained 67 countries. Subregions were then grouped into one of three regions: tropical Americas region (Central America, South America, and the Caribbean), Africa (Eastern Africa, Middle Africa, Western Africa, and Southern Africa), and Asia (Southern Asia and Southeastern Asia). The global data were subset into the three regions for a total of four datasets (Table A1).

Malaria incidence forecasting was performed using the R package “Fable” [64]. To apply the best model specification, both exponential smoothing (ES) and autoregressive integrated moving average (ARIMA) were calculated and averaged, then applied to the forecasting model. Malaria incidence rates at the global and three regional scales were forecasted five years out.

Although care was given to the varying scales of the selected variables, such as preferring variables presented as percentages or “per person” measurements, ultimately, all independent variables were standardized (mean = 0, std = 1) to account for the scale inconsistencies and aid in normalization of the data, as well as aid in the comparison between variable and model comparison and selection. Model coefficients were reported as both standardized and unstandardized. All values were limited to measurements within the country’s boundaries. A correlation exists within each location (country) due to the repeated yearly measures, which are accounted for in the variable “year”. To understand how regionality may affect an independent factor’s influence on malaria incidence, the variable selection was performed using Least Absolute Shrinkage and Selection Operator (LASSO) with R package “glmmLasso” [65] with a gaussian distribution on all four datasets (global, tropical Americas, Asia, and Africa). To find the optimal lambda, candidate lambdas were looped in until the best model was found using Schwarz’s Bayesian Information Criterion (BIC). Significant variables were then fitted to a linear mixed-effects model with R package “lme4” [66]. Country acted as the random effect in the model, with year and the significant landscape, socioeconomic, and economic variables as the fixed effects.

To assess model fit and prediction power, we employed two methods appropriate for linear mixed-effect models. First, we evaluated the coefficient of determination (R^2^) using Nakagawa’s marginal and conditional R^2^ [67]. Marginal R^2^ represents the variance of the fixed effects, while the conditional R^2^ represents the variance of both the fixed and randoms effects. Second, we performed k-fold cross validation to assess the models’ performance. A common concern when using k-fold cross validation with mixed effect models is the inaccurate division of the random effect data into the training folds. The R package “hetoolkit” contains the function “model_cv” that accounts for the random grouping factor by ensuring that the observations for each group are split as evenly as possible across the k-folds [68].

All statistical analyses were conducted using R ver. 4.3.3, developed by the CRAN team at the Vienna University of Economics and Business in Vienna, Austria, and RStudio ver. 2023.12.1+402, developed by Posit Software, PBC in Boston, MA, USA.

## 3. Results

Malaria incidence and trends were examined in 67 countries from years 2000 to 2019 (Table A1). Through the twenty years, malaria incidence per 100,000 population across all countries ranged from 0.16 to 58,908.94 with a mean of 15,030.43 (SD: 16,810.25) (Table A2). Average malaria incidence is highest in the African region (26,392.56; SD: 14,941.43) (Table A2), followed by Asia (1647.82; SD: 3576.21) and the tropical Americas (891.35; SD: 1787.62). Similarly, landscape and socioeconomic parameters also range between regions, with the tropical Americas region retaining the largest average amount of forest cover (52.47; SD: 21.09), average GDP per capita (4576.80; SD: 3112.49), average universal health coverage index (62.67; SD: 11.78), and average tree cover loss (263,266; SD: 741,793) (Table A2). Africa had the highest average malaria spending (3.23; SD: 2.66); agricultural land as a percent of land cover (44.08; SD: 19.63); agricultural, forestry, and fishing value added as percentage of GDP (24.04; SD: 11.27); total natural resources rent as percentage of GDP (8.99; SD: 9.07); and average proportion of Terrestrial KBAs covered by protected areas (51.92; SD: 23.12) (Table A2). Finally, Asia had the largest amount of total official development assistance for biodiversity (65.32; SD: 114.75) (Table A2).

Malaria incidence yearly data were graphed at the global and regional scales; in addition, malaria incidence was forecasted five years into the future (Figure 1). Between 2000 and 2019, total global malaria incidence fell from 1,350,169.30 new cases to 744,711.80; according to the forecasting model, it is projected malaria incidence will continue to fall to possibly 631,185.40 (ES), 547,060.10 (ARIMA), or 589,122.80 (model-averaged) by 2025 at *p* ≤ 0.05. At a regional level, a similar decreasing trend was observed from 2000 (tropical Americas: 29,383.10; Africa: 1,278,910.17; Asia: 41,876.09) to 2019 (tropical Americas: 6075.88; Africa: 725,039.98; Asia: 13,595.91). The tropical Americas were projected to fall to possibly 3818.63 (ES), 2877.40 (ARIMA), or 3348.017 (averaged model) by 2025 at *p* ≤ 0.05. Although Africa has a significantly larger malaria incidence than the other regions, it too was predicted to decrease to 617,530.277 (ES), 549,187.75 (ARIMA), or 583,359.01 (model-averaged) by 2025 (*p* ≤ 0.05). Asia was projected to decrease by 2025 to (*p* ≤ 0.05) 9891.90 (ES), 1514.21 (ARIMA), or 5703.06 (averaged model) (*p* ≤ 0.05) of malaria incidence.

LASSO selection was performed on the independent variables for the four datasets: global, tropical Americas, Africa, and Asia. Insignificant variables from LASSO selection were not included in the linear mixed effects models (Table 2). Our LASSO model computed all variables from the global dataset to be significant (optimal lambada from a minimized BIC = 1668) and was therefore included in the linear mixed effects model. In the tropical Americas LASSO model (optimal lambda = 100,000), year; forest area percent; % of KBAs covered by protected areas; universal health coverage index; agriculture land %; agriculture, forestry, and fishing % of GDP; GDP per capita; natural resource rent %; net ODA received; and malaria spending were found as significant variables and kept for further analysis. The Asia LASSO model (optimal lambda = 1668.101) kept year; forest area percent; % of KBAs covered by protected areas; Universal Health Coverage index; agriculture land %; agriculture, forestry, and fishing % of GDP; natural resource rent %; net ODA received; developmental assistance for biodiversity; malaria spending; and tree cover loss as significant variables. The Africa LASSO model (optimal lambda = 100,000) identified the least number of variables as significant: forest area percent; % of KBAs covered by protected areas; universal health coverage index; agriculture land %; agriculture, forestry, and fishing % of GDP; GDP per capita; natural resource rent; malaria spending; and tree cover loss. Despite being insignificant, the variable year was kept in the Africa dataset to capture the longitudinal aspect of the data.

Significant variables identified from the LASSO selection were fit into a linear mixed effects model, where unstandardized (Table 2) and standardized (Table A3) coefficient values, direction, and significance were recorded (Figure 2 and Figure 3). For the global model, the variables year, GDP per capita, and malaria spending per person were significant (*p* ≤ 0.05), yet the fixed effects explained only 4.3% of the variance (marginal R^2^: 0.043; conditional R^2^: 0.955) (Table 2 and Table A3). Year had the largest magnitude of effect on malaria incidence (β = −2372.18, *p* < 0.001), with a negative relationship showing that malaria incidence decreases by −410.94 incidence for each subsequent year (Table 2). Malaria spending also showed a large, negative relationship with malaria incidence, with each USD 1 of total malaria spending per person decreasing malaria incidence by −651.97 (*p* < 0.001, β = −1607.94.) in the global model. The only significant variable to indicate a positive relationship was GDP per capita (b = 0.56, *p* < 0.001, β = 1585.97). The average prediction error rate for the global model is 0.260 and is the second-best for model accuracy (NRMSE max–min = 0.066).

For the tropical Americas model, % of KBAs covered by protected areas, universal health coverage index; agricultural, forestry, and fishing % of GDP; GDP per capita; natural resources rent percentage of GDP; and net ODA received % of GNI were significant at *p*-value ≤ 0.05 (Figure 2 and Figure 3). In addition, the fixed effects explained 36.9% of the variance (marginal R^2^: 0.369; conditional R^2^: 0.882) (Table 2). Agricultural, forestry, and fishing % of GDP had the strongest positive significant relationship with malaria incidence in the tropical Americas region, with every one percent increasing malaria incidence by 75.10 (β = 971.14, *p* = 0.005). The percentage of KBAs covered by protected areas also showed a strong positive relationship, and a 1% increase in KBA covered by protected areas increases malaria incidence by 21.49 (β = 496.71, *p* < 0.001). Net ODA received was also positively associated with malaria incidence (b = 80.55, *p* < 0.001, β = 625.44) (Table 2 and Table A3). The two economic variables, natural resource rent and GDP per capita, negatively influence malaria incidence, leading to a decrease in malaria incidence by 30.72 for every one percent of total natural resources (*p* = 0.037, β = −321.78) and a decrease of 0.10 for every USD 1 of GDP per capita (*p* = 0.003, β = −276.92) (Table 2 and Table A3). Finally, an increase in the universal health coverage index decreases the malaria incidence rate (b = 35.88, *p* = 0.013, β = −561.60) (Table 2 and Table A3). The tropical Americas model, unfortunately, held an 86.01% prediction error rate when tested using k-fold cross validation and was the worst-performing of the group (NRMSE max–min = 0.0814).

Year, forest area percent, universal health coverage index, and tree cover loss were significant (*p* ≤ 0.05) in the Africa model, with a marginal R^2^ of 0.209 (conditional R^2^: 0.908) (Table 2, Figure 2 and Figure 3). The variable universal health coverage index had the strongest magnitude out of any variable in all of the four models (β = −6196.58, *p* < 0.001) (Table A3), with a one-point increase in universal health coverage index decreasing malaria incidence by 395.92, indicating that health coverage meaningfully minimizes malaria risk in Africa (Table 2). In addition, tree cover loss is strongly associated with decreased malaria rates (b = 0.95, *p* < 0.001, β = −4037.73) (Table 2 and Table A3). Finally, the data show that malaria incidence has significantly fallen over time (b = −227,43, *p* = 0.048, β = −1312.84) (Table 2 and Table A3). The only positive relationship with malaria incidence in the Africa model is forest area percent, with each one percent of forest area leading to an increase of 215.94 malaria incidence (*p* =0.012, β = 5373.18) (Table 2 and Table A3). The model for the African region of the globe had the best predictive power: the percentage error rate was 17.4%, and it was one of the best-performing models (NRMSE max–min = 0.078).

Finally, the Asia model identified variables year, % of KBAs covered by protected areas; universal health coverage index; agriculture, forestry, and fishery % of GDP; natural resource rent; net ODA received as a percent of GNI; and malaria spending per person at 0.05 significance (Figure 2 and Figure 3). The model explained 16.3% of the variance (marginal R^2^: 0.163; conditional R^2^: 0.976) (Table 2). Malaria spending per person was the most impactful variable, with each USD 1 spent decreasing malaria incidence by 874.04 (*p* < 0.001, β = −2155.62) (Table A3). The model also showed that malaria incidence has significantly reduced over time (b = −187.51, *p* < 0.001, β = −1082.43) (Table 2 and Table A3). Counterintuitively, the universal health coverage index is positively related to increased malaria in the Asia region of the globe (b = 80.89, *p* < 0.001, β = 1266.02) (Table 2 and Table A3). Percentage of KBA covered by protected areas (b = 72.63, *p* < 0.001, β =1679.20); natural resource rent (b = 49.29, *p* = 0.026, β = 516.37); and agriculture, forestry, and fishing as a percent of GDP (64.55, *p* < 0.001, β = 834.67) also significantly increased malaria incidence (Table 2 and Table A3). The Asia model, when tested with k-fold cross validation, had a percent error rate of 53.3%, yet performed the best when evaluating the normalized RMSE using the malaria incidence minimum subtracted by maximum (0.043).

Each model shows distinct differences, with no one variable considered significant for every model. This demonstrates that the environment and socioeconomic characteristics of a region vary and that specific characteristics may influence malaria incidence more in one region than in another. The year variable, when significant, collaborated with our forecast (Figure 1) that malaria incidence is decreasing over time. Total official development assistance for biodiversity by recipient countries was the most frequently removed from the LASSO selection, being absent in both the tropical Americas and Africa models and non-significant in the global and Asia models. Agricultural land as a percentage of total land area was never considered significant in any model.

## 4. Discussion

Our goal was to examine the current trend of malaria incidence across the globe and examine its relationship to forest loss and land use change, as well as identify possible key factors that may impact malaria risk at both the global and regional levels. Our results demonstrate that malaria has been and will continue to decrease in prevalence in tropical forested countries through the near future. In addition, modeling malaria incidence against various landscape and socioeconomic variables has highlighted significant differences between regions, indicating that in order to continue to minimize the risk of malaria on global health, regionality must be considered in government policy-making and other intervention methodologies.

From 2000 to 2019, malaria incidence has steadily decreased on both global and regional scales in tropical forested countries (Figure 1). Global malaria has fallen 44.8%, while the tropical Americas, Africa, and Asia have decreased by 79.3%, 43.3%, and 67.3%, respectively. Forecasting using the ES and ARIMA average predicts that global malaria will fall an additional 28.9%, tropical Americas 52.6%, Africa 19.5%, and Asia 58.1% from 2019 to 2025. The trend over the past twenty years is a testament to the joint efforts of governments, non-profits, and individuals in slowing the incidence of malaria on a global scale through new policies, medical treatments, and preventive techniques. However, the world has continued to rapidly change with new threats to both the climate and landscapes that inhabit the tropical regions of the world [69,70]. In a potential domino effect, changes to the environment can impair livelihoods and health and, therefore, immobilize economic status for both individuals and countries; countries with higher cases of malaria experience slower economic growth than countries with lower incidence [71]. Broad policies may or may not be successful in individual regions around the world as each country brings its own history and regional challenges; therefore, it would be advantageous to identify patterns of influence on malaria risk at both the global and regional levels.

Malaria, similar to other infectious diseases, has been linked to a multitude of different landscapes and socioeconomic factors that can affect risk to populations. The variables selected for the study are broad interpretations of potential influences on malaria incidence, such as forest cover and quality, healthcare and socioeconomic status for citizens, and government commitment to malaria elimination. Although more focused datasets created through fieldwork or complimentary projects could target individual influences on malaria incidence more successfully, it was of interest to determine the strength of the models composed of only readily available datasets found through databases and other online sources, many of which are continually updated by international research and governing agencies. However, due to the unspecific nature of the public datasets to direct malaria incidence, the LASSO variable selection was an attempt to specify key components of malaria incidence at both a global and regional scale and to further model to capture the greatest influence at these varying spatiotemporal levels.

Our models showed varying success in capturing malaria incidence risk. Working with numerous ecological variables for 67 different countries made describing broad trends incredibly difficult. The tropical Americas model had the largest marginal R^2^ at 0.369, which can be considered a poor fit when compared to the suggested guidelines [72]. Testing the models using k-fold cross validation indicated that some models, particularly the global and African models, were adequate in prediction with a minimal error rate of 26.0% and 17.4%, respectively (Table 2). The difference between the unsatisfactory coefficient of determination and satisfactory NRMSE may be an indication that the model makes it difficult to fit the existing data yet is still proficient at predicting the values of new data. Tropical Americas had the highest error rate at 86.0% (Table 2) despite also having the highest marginal R^2^, which may lead credence to the opposite: that this particular model fits the data better but is not quite as adept at the prediction of new values. However, the complexity of ecological data may prevent models from reaching higher coefficients of determination or better prediction power, and therefore, it may be wise to explore the novel relationships presented by the models [73].

One goal of the study was to evaluate how deforestation and land use change can impact malaria incidence both globally and regionally. Forest area as a percent of land area, agricultural land as a percent of land area, and tree cover loss were used to explore this relationship, while the average proportion of terrestrial key biodiversity areas covered by protected areas was an attempt to evaluate forest quality [74], or locations within countries with untouched forest. Agricultural land as a percentage of total land area was not significant in any of our models and, therefore, was not evaluated further for interpretation (Table 2. Forest area and tree cover loss were only significant in the Africa model; however, it was interpreted that increased forest area and minimized forest loss are associated with increased malaria incidence (forest area: b = 215.94, *p* = 0.012, β = 5373.18; tree cover loss: b = −0.95, *p* < 0.001, β = −4037.73) (Table 2 and Table A3). Although many studies have concluded that malaria incidence is increased by deforestation, especially in the Amazon or Asia [75,76,77,78,79], a few have indicated that the relationship is not always so clear, especially in African regions: one study found that childhood malaria was generally associated with complete forest cover [80], while another could not find that malaria was associated with deforestation or intermediate forest cover in 17 different African countries [38]. The reason for this relationship has multiple possibilities. First, *Anopheles nili* is a deep or closed-forest mosquito species and is considered one of the largest contributors of malaria in Africa due to its effectiveness and wide range [81]. Although not as prolific as other non-forest species on the continent, there are plenty of deep forests that cover large expanses and house numerous communities; in Central Africa, there is 1.4 million km^2^ of forest under malaria risk, homing a population of 18.7 million people [82]. The amount of forested areas and large populations in malaria-risk areas may be enough to influence the model to assign a positive association between forest areas or areas with low tree cover loss. Another confounding factor is that there can be large variations in malaria incidence within a country. By using country-level data only, in-country regional differences in malaria incidence and trends, as well as distribution of forest in comparison to communities, can get lost. It is possible that a different relationship between forest cover and malaria risk could exist if examining a finer regional scale, but is altered at the country-level. Finally, forests are often areas of high biodiversity; it is not yet fully understood how issues such as hidden reservoirs or disease transmission dynamics may be impacted by regions of high species diversity [83].

This was explored in our model by examining the average proportion of terrestrial key biodiversity areas covered by protected areas. The variable was significant and positively associated with malaria incidence in both the tropical Americas (b = 21.49, *p* < 0.001, β = 496.71) and Asia model (b = 72.63, *p* < 0.001, β = 1679.20) (Table 2 and Table A3). Multiple studies have shown that high biodiversity regions or areas under protection can positively influence malaria incidence [84,85,86], and our models seem to echo a similar finding. Despite the possible risk of malaria incidence, forest conservation and protection of biodiversity hotspots have many other positive benefits that cannot be ignored. They provide key ecosystem services like carbon sequestration, water and soil regulating services, maintenance of ecosystems, and production of natural resources [87,88,89]; they also can mitigate risk for many other zoonotic or infectious diseases and play key roles in the health of human populations [90]. Therefore, the answer is not an increase in the loss of key habitats to minimize malaria risk but a more complete approach to conservation that considers disease ecology and other potential harms when developing and managing protected areas [91].

Healthcare is one of the most direct ways in which the malaria burden can be reduced in populations. Malaria spending per person and the universal healthcare coverage index were two variables included in our models to examine their direct impact on malaria incidence. Malaria spending per person was significant in two of our models, with an increase of USD 1 per person decreasing malaria incidence by 651.97 (*p* < 0.001) and 874.04 (*p* < 0.001) in the global and Asia models, respectively (Table 2). Health expenditures by governments have shown to be beneficial, if not significant, in select African countries [92]. In addition, universal health coverage was demonstrated to strongly reduce the burden of acute illnesses [93]. However, the Asia model had the unique case of universal healthcare coverage being positively associated with malaria incidence (b = 80.89, *p* < 0.001, β = 1266.02) (Table 2 and Table A3). A similar issue was seen in a paper commenting on Zika in countries with or without universal healthcare coverage; although Zika was largely seen in countries with no universal coverage, a few universal health coverage countries had higher Zika incidence than those without. In those situations, it may be that the increased coverage had led to more accurate reporting and, therefore, a higher incidence [94]. Further studies in Asia examining healthcare coverage and spending on malaria incidence may be warranted.

Finally, our models examined several economic variables to capture the effect industries, financial growth, and foreign assistance may have on malaria risk. GDP per capita was significant for both the global (b = 0.56, *p* < 0.001, β = 1585.97) and tropical Americas model (b = −10, *p* = 0.003, β = −276.92) (Table 2 and Table A3). GDP has already been shown to be negatively impacted by malaria, with one study in Tanzania finding malaria burden equaling 1.1% of the GDP [95]. In assessing how GDP impacts malaria, we were surprised to find a positive correlation in the global model despite the linear prediction plots (Figure 2) identifying a positive relationship. We believe that variation in levels of malaria risk between countries caused GDP to behave like a constant in the global model; in fact, when examined regionally in the Tropical Americas, we see the negative relationship expected. Net ODA received was also significant in both the tropical Americas model (b = 80.55, *p* < 0.001, β = 625.44) and the Asia model (b = 271.27, *p* < 0.001, β = 2106.21) (Table 2 and Table A3). The variable was not specific to aid given for malaria or health but general promotion for economic development and welfare in countries and territories; therefore, it reflects the countries with a need for aid stimulation. Another paper, although focused specifically on development assistance for health and disease burden, also found a positive correlation between ODA and malaria risk [96]. The apparent indication is that countries in most need of aid may be those with the strongest disease burden. Two variables were focused on economic outputs and were an attempt to capture countries’ interactions with natural areas: total natural resource rents as a percent of GDP and agriculture, forestry, and fishing as a percent of GDP. The latter was significant for both the tropical Americas (b = −30.72, *p* = 0.037, β = 971.14) and Asia (b = 64.55, *p* < 0.001, β = 834.67) models with a positive association with malaria risk (Table 2 and Table A3). A previous study had previously linked malaria to deforestation-related commodities, including timber, cocoa, coffee, and other natural products, with approximately a 20% increase in malaria incidence in areas of deforestation attributed to the exportation of these products [97]. Although our variable contained additional exportation revenues, including fishing, it may still be an indication that our models are capturing the increased malaria risk from natural resource-based industries and driving incidence in these regions. Lastly, natural resources rent, like net ODA, was significant in the tropical Americas (b = −30.72, *p* = 0.037, β = −321.78) and Asia (b = 49.29, *p* < 0.001, β = 516.37) (Table 2 and Table A3). The relationship between resource-rich countries and human well-being is contradictory among papers; Lyatuu et. al. found that the short-term effects of natural resource rent incomes are correlated to increased life expectancies within sub-Saharan Africa and that the increased revenue results in additional revenue for governments for new policies and programs [98]. However, Chang and Wei discovered the opposite when examining malaria especially, and that resource-rich countries are more highly associated with malaria cases of infection and death, attributed to reluctance by governments to reinvest in national programs and poor conditions in mining and drilling areas [99]. Our models indicate a positive relationship between natural resource rents and malaria incidence in Asia but a negative relationship in the tropical Americas, suggesting that the impact may be region-specific. Whether the increase revenue from natural resources helps or hinders malaria incidence may be dependent on how the government reinvests that income, and that varies by each country.

We experienced limitations and challenges when creating our models. Many of our dependent and independent variables violated the assumption of normality even after standardization and showed imperfection with the linearity of residuals and, therefore, may not have been the best fit for a linear mixed model. Improvement in model fit and explanation of variance for both global and regional datasets may occur if other modeling methods are implemented. In addition, a few countries had more missing data than others, especially in the early years of agriculture, forestry, and fishing, which valued an added percent of GDP for Djibouti. The missing data were added through the “first observation carried backwards” methodology. We felt it was important to include the agricultural, forestry, and fishing percent of GDP variable and not remove it from the model due to one country. In addition, a more detailed analysis of the data indicated there was very little variation in value for the years available (0.69%), and considering the missing data were the most historic, we decided that “first observation carried backwards” would be an appropriate solution.

In addition, the choice of using country-level spatial data introduced an intrinsic set of limitations on how results can be interpreted by individual countries or regions within countries. The goal of the study was to determine the broad trends of malaria across vast geographical regions, whether these regions carry their own unique risk of increased malaria incidence, and how these regions compare to a global view of malaria incidence. However, our selected variables may vary greatly across an individual country, and areas of high malaria incidence may not always correlate to the country-wide data found for each of our proposed risk factors. Moreover, we attempted to best capture vulnerabilities for populations to malaria through a varied selection of landscape and socioeconomic variables; however, specific variables related to social inequalities were not included due to data unavailability for all the countries at the level we wanted. This problem was compounded by the use of country-level data, which may limit insight into the in-country regions most at risk for malaria due to these inequalities. Increased malaria incidence has an established link to social inequalities [100,101,102], and not having more detailed variables exploring that relationship may lead our results to miss a key component of malaria incidence in certain countries and, therefore, greater geographical regions versus others.

Another failing of our model is the exclusion of information on *Plasmodium* sp. in reported malaria cases. *Plasmodium knowlesi* is a zoonotic malaria parasite primarily found in Southeastern Asia [103] and may be influenced by different landscape, socioeconomic, and economic risk factors compared to human malaria [104]. Further studies on regionality for malaria risk should consider zoonotic malaria cases in comparison to the other forms of human malaria to possibly highlight emerging risk factors that could increase *p. knowlesi* incidence.

Finally, although our study data and analysis focused on the period pre-SARS-CoV-2 global pandemic, the lack of inclusion of the pandemic may have led to inaccurate malaria projection in our forecasting model or incomplete risk factors for our global and regional linear mixed effects models. The COVID-19 pandemic impacted healthcare and government systems, resulting in indirect increases in malaria incidence, particularly in African countries and other malaria-endemic countries [105,106,107]. These causes range from interruptions to healthcare systems, including delayed diagnosis or treatment, to disruption in preventative and control measures [108]. Our forecast does not include information about the pandemic in its model, nor do our global and regional linear mixed-effects models account for the effects of the COVID-19 pandemic on tested risk factors and their subsequent models. However, there is increased concern about new zoonotic disease spillover and pandemics from land-use change, decreasing ecosystem health, and deforestation [109,110]. The same risk factors for increased malaria incidence may also increase the risk of new pandemics; therefore, it is worthwhile to determine how malaria incidence may fluctuate in the future with more epidemics or pandemics. A prospective research aim would be to determine the extent of the discrepancy between the malaria forecasts and actual malaria incidence numbers post-pandemic and discover the best method to capture potential increases in indirect disease incidence during a global health crisis. In addition, the risk models can be run using new risk factors related to vulnerability to COVID-19 and determine which countries may be more at risk for future pandemics coupled with increased endemic disease incidence due to their health or government systems.

## 5. Conclusions

Malaria cases around the globe are decreasing, but as the world continues to experience increased deforestation and land use change, it is unclear what relationship it will have on malaria incidence in tropical forest countries. Through variable selection and linear mixed effect models, we found that regionality plays an important role in influencing forest loss, land use change, and socioeconomic factors on malaria incidence. However, low model fit or predictive accuracy for some of our models indicates that further refinement can be accomplished. Our models also highlighted key areas of future research, including the relationship of forest cover to malaria risk in Africa, an exploration into the effects of universal health coverage on disease reporting to better evaluate program or healthcare policies, the potential negative impacts of increased disease risk with some conservation policies, and how best to negate the ill-effects to maximize the positive benefits of ecosystem services that come from protected areas.

## Figures and Tables

**Figure 1 ijerph-21-00576-f001:**
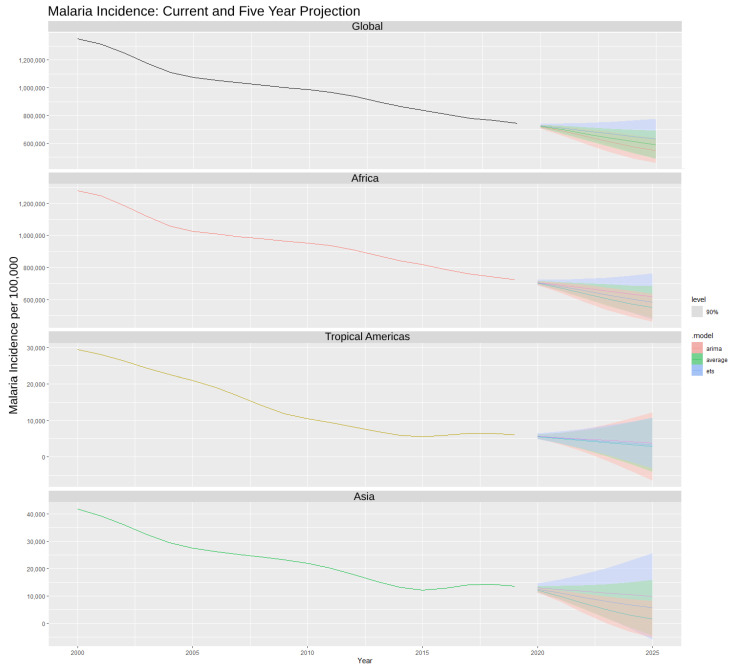
Actual malaria incidence per 100,000 population for the years 2000 to 2019 and projected malaria incidence from years 2020 to 2025 using models exponential smoothing, ARIMA, and an averaged model of both at *p*-value ≤ 0.05 and confidence interval of 90%. Forecasts are divided by global and the three regions: Africa, tropical Americas, and Asia.

**Figure 2 ijerph-21-00576-f002:**
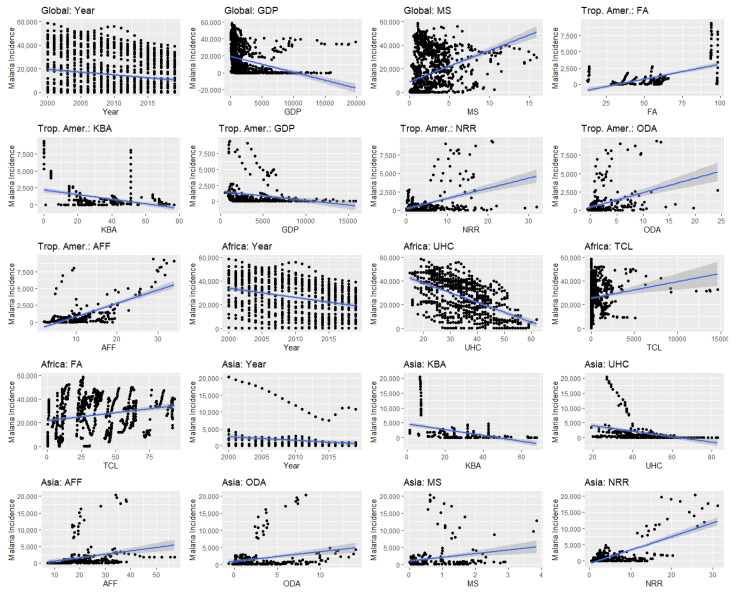
Linear model prediction plots for each of the significant variables found from the linear general mixed global and regional models. The regression line is highlighted in blue. GDP = GDP per capita; MS = malaria spending per person; FA = forest area (% of land area); AGB = above-ground biomass in forest; KBA = average proportion of terrestrial key biodiversity areas covered by protected areas (%); NRR = total natural resources rent (% of GDP); ODA = net ODA received (% of GNI); UHC = universal health coverage service index; TCL = country tree cover loss; AFF = agricultural, forestry, and fishing (% of GDP).

**Figure 3 ijerph-21-00576-f003:**
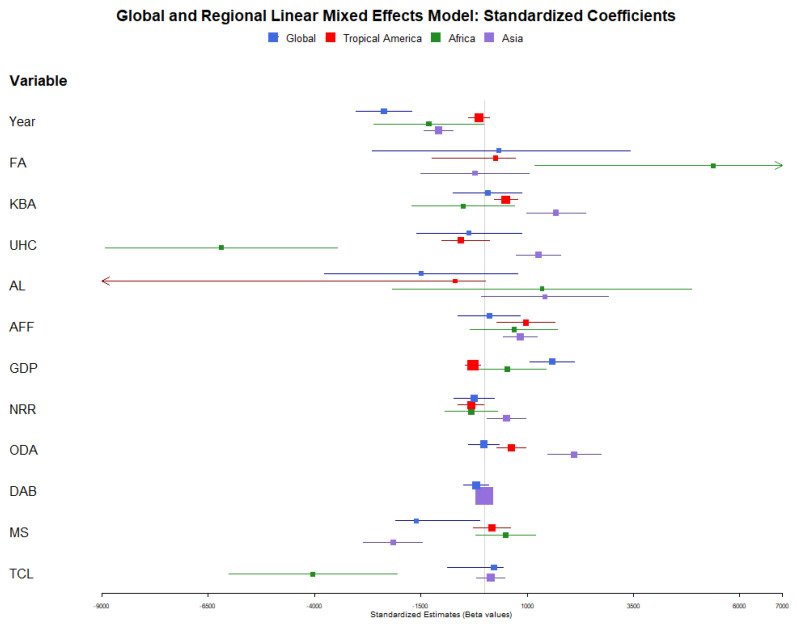
Forest plot of the standardized model estimates for each variable of the global, tropical Americas, Africa, and Asia linear mixed effects model. Colored squares indicate the mean estimate, and the line is the confidence interval. FA = forest area (% of land area); KBA = average proportion of terrestrial key biodiversity areas covered by protected areas (%); UHC = universal health coverage service index; AL = agricultural land (% of land area); AFF = agricultural, forestry, and fishing (% of GDP); GDP = GDP per capita; NRR = total natural resources rent (% of GDP); ODA = net ODA received (% of GNI); DAB = total official development assistance for bio-diversity, by recipient countries (millions of constant 2020 USD); MS = malaria spending per person; TCL = country tree cover loss.

**Table 1 ijerph-21-00576-t001:** Variable names, year available from online database, definitions adapted from online sources, and citation for online source.

Name	Years	Definitions	Citation
Malaria Incidence (per 100,000 population)	2000–2019	The number of new cases in a year divided by the mid-year population size; per 100,000 population.	[52]
Total Malaria Spending per Person (constant 2019 USD$)	2000–2017	Total malaria spending (government, out-of-pocket, prepaid private) per person (constant 2019 United States Dollars)	[53]
Agricultural land (% of land area)	2000–2019	Agricultural land refers to the share of land area that is arable, under permanent crops, and under permanent pastures.	[54]
Agricultural, forestry, and fishing, value added (% of GDP)	2000–2019	Agriculture, forestry, and fishing; includes forestry, hunting, and fishing, as well as cultivation of crops and livestock production. Value added is the net output of a sector after adding up all outputs and subtracting intermediate inputs. It is calculated without making deductions for depreciation of fabricated assets or depletion and degradation of natural resources.	[55]
Forest area (% of land area)	2000–2019	Forest area is land under natural or planted stands of trees of at least 5 m in situ, whether productive or not, and excludes tree stands in agricultural production systems (for example, in fruit plantations and agroforestry systems) and trees in urban parks and gardens.	[56]
GDP per capita (current USD)	2000–2019	GDP per capita is gross domestic product divided by midyear population. GDP is the sum of gross value added by all resident producers in the economy plus any product taxes and minus any subsidies not included in the value of the products. It is calculated without making deductions for the depreciation of fabricated assets or for the depletion and degradation of natural resources.	[57]
Total natural resources rent (% of GDP)	2000–2019	Total natural resources rents are the sum of oil rents, natural gas rents, coal rents (hard and soft), mineral rents, and forest rents.	[58]
Net ODA received (% of GNI)	2000–2019	Net official development assistance is disbursement flows (net of repayment of principal) that meet the DAC definition of ODA and are made to countries and territories on the DAC list of aid recipients.	[59]
Universal health coverage (UHC) service coverage index	2000, 2005, 2010, 2015–2019	Coverage of essential health services (defined as the average coverage of essential services based on tracer interventions that include reproductive, maternal, newborn and child health, infectious diseases, non-communicable diseases, and service capacity and access among the general and the most disadvantaged population). The indicator is an index reported on a unitless scale of 0 to 100, which is computed as the geometric mean of 14 tracer indicators of health service coverage.	[60]
Average proportion of Terrestrial Key Biodiversity Areas (KBAs) covered by protected areas (%)	2000–2019	Proportion of important sites for terrestrial biodiversity that are covered by protected areas.	[61]
Total official development assistance for biodiversity by recipient countries (millions of constant 2020 USD)	2002–2019	Official development assistance on conservation and sustainable use of biodiversity, defined as gross disbursements of total Official Development Assistance (ODA) from all donors for biodiversity by recipient country.	[62]
Country tree cover loss (km^2^)	2001–2019	Country tree cover loss: Hectares of tree cover loss at a national level between 2001 and 2021. Tree cover is defined as all vegetation greater than 5 m in height and may take the form of natural forests or plantations across a range of canopy densities. “Loss” indicates the removal or mortality of tree cover categorized by percent canopy cover in 2000 (≥30% threshold) and can be due to a variety of factors, including mechanical harvesting, fire, disease, or storm damage. As such, “loss” does not equate to deforestation.	[10] Global Administrative Areas Database, version 3.6. Available at http://gadm.org/ (accessed 12 Janauary 2023)

**Table 2 ijerph-21-00576-t002:** Linear mixed effect model with malaria incidence as the dependent variable and country as the random effect. The table reports the model estimates as unstandardized (b). Bolded p-values indicate significance at *p* ≤ 0.05. Missing values indicate that the variable was not significant during the LASSO variable selection. Root mean squared error (RMSE) and normalized root mean square error (NRMSE) values were generated from 10 k-fold cross validation. FA = forest area (% of land area); KBA = average proportion of terrestrial key biodiversity areas covered by protected areas (%); UHC = universal health coverage service index; AL = agricultural land (% of land area); AFF = agricultural, forestry, and fishing (% of GDP); GDP = GDP per capita; NRR = total natural resources rent (% of GDP); ODA = net ODA received (% of GNI); DAB = total official development assistance for biodiversity, by recipient countries (millions of constant 2020 USD); MS = malaria spending per person; TCL = country tree cover loss.

	GLOBAL			TROPICAL AMERICAS			AFRICA			ASIA		
	Estimates	CI	*p*	Estimates	CI	*p*	Estimates	CI	*p*	Estimates	CI	*p*
(Intercept)	8.44 × 10^5^	6.170 × 10^5^–1.072 × 10^6^	**<0.001**	5.10 × 10^4^	3.33 × 10^4^–1.35 × 10^5^	**<0.001**	4.86 × 10^5^	3.82 × 1^4^–9.34 × 10^5^	**0.033**	3.69 × 10^5^	2.51 × 10^5^–4.86 × 10^5^	**<0.001**
Year	−410.94	−524.82–−297.07	**<0.001**	−23.61	−65.71–18.50	0.271	−227.43	−452.93–−1.93	**0.048**	−187.51	−246.65–−128.38	**<0.001**
FA	15.79	−106.08–137.67	0.799	−10.32	−49.60–28.97	0.606	215.94	47.65–384.22	**0.012**	−9.29	−60.65–42.07	0.722
KBA	2.91	−32.29–38.10	0.871	21.49	9.65–33.32	**<0.001**	−22.04	−74.28–30.20	0.408	72.63	42.66–102.61	**<0.001**
UHC	−23.50	−102.47–55.47	0.559	−35.88	−63.99–−7.77	**0.013**	−395.92	−570.36–−221.48	**<0.001**	80.89	47.70−114.08	**<0.001**
AL	−74.67	−188.79–39.44	0.199	−34.76	−70.64–1.11	0.057	67.65	−109.26–244.56	0.453	71.18	−4.05–146.41	0.064
AFF	8.41	−48.60–65.43	0.772	75.10	22.49–127.72	**0.005**	53.52	−25.94–132.99	0.186	64.55	33.16–95.94	**<0.001**
GDP	0.56	0.37–0.75	**<0.001**	−0.10	−0.16–−0.03	**0.003**	0.19	−0.13–0.51	0.250	-	-	-
NRR	−23.51	−68.72–21.70	0.308	−30.72	−59.58–−1.85	**0.037**	−29.56	−88.84–29.73	0.328	49.29	6.02–92.56	**0.026**
ODA	−2.55	−49.02–43.92	0.914	80.55	36.23–124.88	**<0.001**	-	-	-	271.27	190.51–352.02	**<0.001**
DAB	−2.89	−7.03–1.25	0.171	-	-	-	-	-	-	−0.18	−1.47–1.12	0.787
MS	−651.97	−851.51–−452.43	**<0.001**	69.31	−111.13–249.76	0.450	198.99	−86.52–484.51	0.172	−874.04	−1157.68–−590.39	**<0.001**
TCL	−0.05	−0.21–0.10	0.509	-	-	-	−0.95	−1.42–−0.49	**<0.001**	0.03	−0.04–0.11	0.398
** *Random Effects* **												
σ2		1.472 × 10^7^		σ2		5.329 × 10^5^		σ2		2.038 × 10^7^	σ2	6.866 × 10^5^
**ICC**		0.95		ICC		0.81		ICC		0.88	ICC	0.97
Marginal R2/Conditional R2		0.043/0.955		Marginal R^2^/Conditional R^2^		0.369/0.882		Marginal R^2^/Conditional R^2^		0.209/0.908	Marginal R^2^/Conditional R^2^	0.163/0.976
** *K-* ** ** *fold Cross Validation* **												
RMSE		3902.758		RMSE		767.204		RMSE		4595.974	RMSE	877.666
NRMSE (RMSE/mean (y))		0.260		NRMSE (RMSE/mean (y))		0.8601		NRMSE (RMSE/mean (y))		0.174	NRMSE (RMSE/mean (y))	0.533
NRMSE (RMSE/y max–y min)		0.066		NRMSE (RMSE/y max–y min)		0.0814		NRMSE (RMSE/y max–y min)		0.078	NRMSE (RMSE/y max–y min)	0.043

## Data Availability

The datasets generated and analyzed during the current study are openly available at: https://osf.io/kn7gb/?view_only=ba06cf02b572488fb697748f9616af28, accessed on 16 March 2024.

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
