# Peer review of "Landscape and Socioeconomic Factors Determine Malaria Incidence in Tropical Forest Countries"

_ijerph, 2024, doi:10.3390/ijerph21050576_

Round 1
Reviewer 1 Report
Comments and Suggestions for Authors
This is a very interesting article in which the authors have attempted to examine current trends in malaria incidence in the face of the impacts of deforestation and land use, and to identify key factors that may influence the risk of malaria transmission at global and regional levels in tropical forest areas.
A very relevant topic given the emerging issue of climate change and global warming.
An innovative approach mentioned by the authors in this article is the fact that the models generated, and the assessment made considering several factors, rather than looking at isolated factors.
Q1. As a general question, it should be noted that all analyses were conducted using malaria incidence data for the period 2000-2019, i.e. before the COVID-19 pandemic.
It is known that the pandemic has changed the malaria incidence around the world, and that there have been huge increases in malaria incidence in several endemic countries in the post-pandemic period. My question is: what impact has the pandemic had on the models generated in the paper?
Corrections/Suggestions
In line 26, add more recent data on the incidence of malaria worldwide.
In line 34, it is necessary to indicate which period is related to the increase in malaria cases worldwide.
In line 40, write the word Plasmodium in italics, with the first letter in capitals.
In line 152, it really does not make sense to keep Cabo Verde, as this country has been certified as a malaria-free country in 2024.
Explain the difference between these two statements:
In lines 193 and 194 “Through the twenty years, global malaria incidence per 100,000 population ranged from 0.16 to 58,908.94 with a mean of 15,030.43 (SD: 16,810.25) and in lines 208 and 209, “Between 2000 to 2019, global malaria incidence fell from 1,350,169.30 new cases to 744,711.80”
In line 225, it is indicated that LASSO model is shown in Table 2, but it is not. In Table 2 are shown the variables selected by LASSO to the linear mixed effect model. This needs to be made clearer.
In line 245, what do RMSE and NRMSE stand for?
In line 255, add “Table 2” after “effects model”
In lines from 253 to 266 there are several comparisons, for example, “Year had the largest magnitude of effect on malaria incidence”. Please, indicate in which Table the β coefficients are shown. Are they standardized? If yes, please, indicate this. If not, comparisons between variables suggesting that one variable has a greater impact than the other are not appropriate. The same for all comparisons regarding the Americas, Africa, and Asia models.
In line 305 is indicated that the variable “FA percent” was at 0.05 significance level for the Asia model, but it is incorrect.
Reviewer 2 Report
Comments and Suggestions for Authors
It is an interesting study that evaluated how some drivers (like deforestation, landscape dynamics, and socioeconomic factors) within the tropical Americas, Africa, and Asia- affect malaria incidence at the global and regional scale. Authors used three different models for the evaluation and a long data series (from 2000-2019) from 67 countries.
To improve the interpretation of the findings of this study, some adjustments need to be made.
Major comments:
To add (as supplementary material), the list of countries included in the study (n=67).
Variables:
More information about the construction of variables should be added in the supplementary materials. It seems that the most part of variables was built using the https://databank.worldbank.org/source/world-developmentindicators, and authors said that all variables were extracted at the country-level scale.
About this topic, some considerations are necessary:
For example, in Tropical America (defined as countries in Central America, South America, and the Caribbean), Brazil is a big country with a high burden of malaria. In this country, 99% of cases occur in the Amazon region. In Brazil there are 215.3 million inhabitants; in the Amazon region there are 38 million. To calculate the annual incidence of cases (MI), what was the denominator used? The entire Brazilian population (n=215.3 million) or the population at risk (n=38 million)?. In the first case, the MI for 2019 would be 72.8/100,000; in the second 412.9/100,000. This will naturally give a different incidence and may bias the results.
Although the same consideration must be made for the other variables, special care must be given to the variable GDP = GDP per capita. Using the example of Brazil again, while the country's GDP in 2019 was 8,845.32, in the Amazon region it was only 5,886.5.
About methods, more information should be added as supplementary material on how each of the three approaches were constructed: exponential smoothing (ES), Autoregressive Integrated Moving Average (ARIMA) and averaged model.
How did the authors deal with regional heterogeneities and deep inequalities within each country?
DISCUSSION:
How can the results found in contexts of profound inequality and heterogeneous scenarios between countries be interpreted?
A better discussion about results must de added in the discussion chapter. To understand the malaria context in each setting is extremely important for interpretation of results. The countries where malaria occur are characterized by profound social inequality. There are places where malaria transmission was interrupted many years ago, while in others the endemic disease is still very important.
The lack of significance of some variables that would apparently be relevant may be due to inequalities existing in malaria-endemic areas in countries and areas where there is no malaria. If the denominator used to construct the variables was national, there may be biases in the interpretation of the results.
Review the list of references. Not all references are well cited.
Minor comments:
Line 27: When generic names are used, it must be added sp. (ex. Anopheles sp.). Anopheles must be written in italic; sp. in non-italic.
Line 28: Please, correct the word; “Plasmodium falciparium”. It is Plasmodium falciparum.
Line 29: to add “and deaths” after illnesses.
Line 49: “infectious plasmodium”; please change plasmodium for Plasmodium (in italic)
Line 289: Change the word “topical” Americas for tropical.
Line 356: Authors said: “ However, the world has continued to rapidly change with new threats to both the climate and land-scapes that inhabit the tropical regions of the world [58, 59]”. As the Covid-19 pandemic has had important effects on the burden of malaria in many countries, this information should be added and discussed here.
Line 369: Authors use the term “malaria eradication”. Governments are commitment to malaria elimination, not eradication; definitions are different.
Line 415: Authors discuss about confounding factors, but a better discussion about using national indicators must be addressed specifically in countries characterized for profound inequalities.
Line 503: Use italic for “Plasmodium” and add sp.
